# Calcium-Sensing Receptor (CaSR), Its Impact on Inflammation and the Consequences on Cardiovascular Health

**DOI:** 10.3390/ijms22052478

**Published:** 2021-03-01

**Authors:** Sai Sahana Sundararaman, Emiel P. C. van der Vorst

**Affiliations:** 1Interdisciplinary Centre for Clinical Research (IZKF), RWTH Aachen University, 52074 Aachen, Germany; ssahanasunda@ukaachen.de; 2Institute for Molecular Cardiovascular Research (IMCAR), RWTH Aachen University, 52074 Aachen, Germany; 3Department of Pathology, Cardiovascular Research Institute Maastricht (CARIM), Maastricht University Medical Centre, 6229 ER Maastricht, The Netherlands; 4Institute for Cardiovascular Prevention (IPEK), Ludwig-Maximilians-University Munich, 80336 Munich, Germany; 5German Centre for Cardiovascular Research (DZHK), partner site Munich Heart Alliance, 80336 Munich, Germany

**Keywords:** calcium-sensing receptor, CaSR, G-protein coupled receptors, inflammation, cardiovascular disorders, atherosclerosis, vascular calcification, myocardial ischemia, adipose tissue, hypertension

## Abstract

The calcium Sensing Receptor (CaSR) is a cell surface receptor belonging to the family of G-protein coupled receptors. CaSR is mainly expressed by parathyroid glands, kidneys, bone, skin, adipose tissue, the gut, the nervous system, and the cardiovascular system. The receptor, as its name implies is involved in sensing calcium fluctuations in the extracellular matrix of cells, thereby having a major impact on the mineral homeostasis in humans. Besides calcium ions, the receptor is also activated by other di- and tri-valent cations, polypeptides, polyamines, antibiotics, calcilytics and calcimimetics, which upon binding induce intracellular signaling pathways. Recent studies have demonstrated that CaSR influences a wide variety of cells and processes that are involved in inflammation, the cardiovascular system, such as vascular calcification, atherosclerosis, myocardial infarction, hypertension, and obesity. Therefore, in this review, the current understanding of the role that CaSR plays in inflammation and its consequences on the cardiovascular system will be highlighted.

## 1. Introduction

The calcium sensing receptor or CaSR, as the name suggests, is a receptor that senses calcium in the extracellular environment of a cell. The human *CASR* gene is located on chromosome 3 and is comprised of 8 exons [1]. Exons 2 to 7 encode for a 1078 amino acid G protein coupled receptor, present on the surface of cells. The receptor was first cloned a bit more than 25 years ago [2] and since then it has been elaborately studied in various pathologies. CaSR is present in various organs and its exact function can differ based on its location. CaSR is not only activated by calcium ions, but by numerous ligands that including various cations, polyamines, polypeptides and antibiotics [3]. The combination of this wide variety of ligands triggers intricate intracellular signaling networks, which eventually influence many physiological and pathological processes.

The role of CaSR in mineral homeostasis has been widely studied due to its presence in calcitropic tissues such as kidneys, bone, and parathyroid glands. However, more recently the role that CaSR plays in the (patho)physiology of other metabolic processes is also being intensively investigated, while the role it plays in cardiovascular health is gaining scientific interest. This is especially fueled by the observation that CaSR is expressed on the surface of vascular and hematopoietic cells, which play a vital role in inflammatory processes and thereby has subsequent repercussions on cardiovascular diseases (CVDs).

A variety of studies have already been performed to investigate the effect of CaSR on inflammation and cardiovascular biology. This review will therefore highlight the major findings to give a comprehensive overview of different signaling pathways activated by CaSR and the pathological consequences of this activation on inflammation and especially in tissues relevant to CVD.

## 2. CaSR Structure and Regulation

The CaSR belongs to the family of G-protein coupled receptors, also known as the seven transmembrane domain receptors. First discovered in 1993, CaSR was the first membrane protein observed to have ion sensing capacities. It can sense even miniscule fluctuations of extracellular calcium in the human body [4]. The receptor is known for its ability to activate different intracellular signaling pathways based on its location and the type of ligand to which it binds to. CaSR is present in a multitude of organs, like parathyroid glands, kidney, bone, skin, the gut, pancreas, lungs, and heart. Concerning the ligands that bind to the receptor, extracellular calcium ions are not the only potent agonist of this receptor. There are a wide range of ions that can act as a ligand for the CaSR and these can act together to activate the receptor and induce intracellular signaling. The CaSR responds to di- and tri-valent cations including Sr^2+^, Ba^2+^, Mg^2+^, Ba^2+^, Mn^2+^, Ni^2+^, Gd^3+^, La^3+^ and Al^3+^ [3,5]. While ions such as Ca^2+^ and Mg^2+^ act as activators, sulphates act as negative allosteric modulators as well as stabilizers for the active confirmation of the receptor [6]. In addition, serum phosphate [7] binding to the receptor is considered necessary to maintain its structural integrity in its inactive state as well as regulate its activity. Polycations also tend to be agonists for which the effectiveness is dependent on the positive charge. These cations also often act together with amino acids to activate the receptor. Apart from ions, CaSR is also activated by polyamines such as spermines and spermidines, polypeptides such as polylysine and polyarginine, amyloid B peptides, glutathione and aminoglycoside antibiotics [3]. Upon binding of a ligand to the receptor, CaSR can induce various cellular processes, such as secretion of hormones, chemotaxis, apoptosis, proliferation and differentiation [8]. These intracellular signaling processes are believed to be dependent on each other, as well as having the ability to influence each other.

The structures of the inactive and active receptor have been well established [6,9] (Figure 1). The functional receptor exists as a disulfide-linked homodimer that is expressed on the cell surface [10]. It consists of three structural domains and has 1078 amino acids: a large N-terminal extracellular domain (ECD) with a cysteine rich domain (CRD) that links the ECD to a seven transmembrane domain, as well as an intracellular C-terminal domain [11]. The ECD, essential for the binding of ligands, is the biggest and has a characteristic shape resembling the Venus flytrap (VFT) and therefore is named accordingly [12]. This domain has two protomers that each contain a bi-lobed VFT that are separated from each other when the receptor is inactive [6]. When the receptor turns active, usually after an L-amino acid binds to the receptor and closes the groove between the lobes, conformational changes to the receptor occur bringing both the protomers in close proximity of each other [13]. In the C-terminal domain, protein kinase A (PKA) and protein kinase C (PKC) phosphorylation sites are present, which when bound to protein kinases, trigger downstream signaling pathways [14,15]. The intracellular domain is made up of different amino acids that help with lodging the receptor on the cell surface and intracellular signaling.

The intracellular signaling by CaSR is expected to occur in different stages (Figure 1). The first stage after receptor activation is the coupling of CaSR to G_i/o_ and G_q/11_ families of heterotrimeric G proteins that consequently activate signaling pathways [16]. The phospholipases are also activated, which in turn activate inositol triphosphate (IP_3_) and diacylglcerol (DAG) pathways. G_i/o_ along and adenylate cyclase (AC) inhibits cyclic adenosine monophosphate (cAMP) production. The inositol phosphate pathway oversees the control of the intracellular calcium levels, especially in the endoplasmic reticulum (ER). The second stage is when the activated DAG and the cytosolic calcium activates rat sarcoma (RAS), mitogen-activated protein kinase (MAPK) and c-Jun n-terminal kinases (JNK) pathways [17], resulting in a wide range of cellular processes. These pathways are also known to influence the receptor function using a positive feedback mechanism. Finally, different ionic and pH conditions and calcimimetics usually tend to activate transcription factors complexes such as nuclear factor kappa-light-chain-enhancer of activated B cells (NF-κB) to control various cellular processes, which have been mentioned earlier and are visualized in Figure 1.

## 3. CaSR in Cardiovascular System-Related Cells

### 3.1. CaSR in Non-Hematopoietic Cardiovascular Cells

Several studies have identified the presence of CaSR on the membrane of vascular cells and indicated that the receptor is an active player regarding the physiology of the cardiovascular system. For the first time, a study by Wang et al. demonstrated the presence of CaSR in cardiac myocytes. It was observed that an increase in the calcium content of the supernatant of the cell culture medium in which rat cardiac myocytes were grown in-vitro also triggered an increase in the intracellular calcium levels and increased cardiac activity [18]. The consistent activation and reactivation of the receptor with the changing calcium levels due to the feedback mechanism helps with the normal contractility of the muscle cells. Therefore, CaSR is not just responsible for regulating the ion channels but also for regulating the membrane potential. Calcium ions enter the cardiac myocytes via L type ion transport channels, sepcially the L type Ca^2+^ channels (LTCC). This leads to increased secretion of Ca^2+^ from the endoplasmic and sarcoplasmic reticulum which in turn leads to an increased contractility of the cells. When cultured cardiomyocytes were stimulated with oxLDL in-vitro to imitate hyperlipidemia, it was seen that the expression of CaSR significantly increases. Additionally, an increased rate of apoptosis and angiogenesis through mitochondrial pathways was observed [19].

Perhaps the most important non-hematopoietic cell type in vascular tissues is the smooth muscle cell. This is the cell type that influences and plays a decisive role in the contraction and relaxation of the vasculature. CaSR is expressed in vascular smooth muscle cells (vSMCs) [20]. Activation of CaSR in aortic SMCs allows calcium to enter the cells via receptor-operated channels (ROC). These channels signal intracellularly through the phospholipase C (PLC)/PKCε pathway resulting in a CaSR mediated rise of the intracellular calcium levels [21]. Mice that lack CaSR in SMCs were observed to have various compromised vasculature functions, such as vessel tone and blood pressure [22]. Impaired vascular contractility was also found to be independent of the endothelium and mainly reliant on vSMC CaSR. Mice with impaired vascular contractility were also observed to be hypotensive [22], further confirming an important role of CaSR in the vascular tone.

Besides SMCs and the cardiomyocytes, endothelial cells (ECs) also play an important role in the cardiovascular system as they line every vessel. For example, EC dysfunction is the start of an inflammatory process in the arteries. CaSR is also expressed in aortic ECs, and it is mainly localized to the cytoplasm rather than the cell membrane [23]. This is believed to be caused by the receptor recycling or very high biosynthesis due to different post-translational modifications. Both mRNA and CaSR proteins were discovered in vascular ECs in rat and rabbit mesenteric arteries, porcine coronary arteries [24] and human aorta [25]. Studies indicate that extracellular calcium induces nitric oxide (^●^NO) production in ECs through the CaSR [26,27].

### 3.2. CaSR in Hematopoietic Cardiovascular Cells

CaSR is expressed on various immune cells, such as monocytes, macrophages, proerythroblasts, erythroblasts and megakaryocytes [28]. Myeloid precursors also express CaSR but to a lesser extent as compared to other bone marrow cells such as proerythroblasts. Furthermore, it has been demonstrated that CaSR is expressed by T lymphocytes [29], although not by B lymphocytes [30].

The presence of CaSR on the surface of hematopoietic cells contributes majorly to cardiovascular biology as for example the expression of CaSR on monocytes has been implicated in chemotaxis, a key process in inflammatory diseases [31]. CaSR present in macrophages also plays a vital role in micropinocytosis, which facilitates pathogen or tissue damage sensing through pattern recognition [32].

The presence of CaSR in peripheral blood monocytic cells was demonstrated by Olszak et al., who confirmed that the receptor was also present on the cell surface. The expression of CaSR is necessary for the chemotactic response by the cells [33].

## 4. Pathological Role of CaSR in Inflammation and Cardiovascular Diseases

Abnormal expression and function of the CaSR gene has been studied intensively suggesting that the receptor plays an important role in not just calcium homeostasis related disorders but also in various other non-calcium related diseases in humans. It is now well established that almost all components of the cardiovascular system functionally express CaSR, and that this receptor seems to be quite important for maintaining vascular integrity. As of now, there have been more than 230 mutations found within the *CASR* gene and more than half of which cause the receptor to become dysfunctional [34]. Moreover, CaSR polymorphisms are known to be associated with coronary artery diseases (CADs) such as myocardial infarction (MI) and atherosclerosis [35]. These abnormalities of CaSR in hematopoietic cells and vascular cells contribute to a variety of conditions, especially the promotion of local and systemic inflammation, which are destructive to vascular tissue and eventually cause various morbidities. The pathological role of CaSR will be discussed in more detail below (Figure 2).

### 4.1. Imbalance in Mineral Homeostasis and Cardiovascular Disease

To support an array of cellular functions, the concentration of mineral ions in the blood must be kept within a certain physiological range [36]. When the concentrations increase or decrease outside this range the equilibrium in the mineral homeostasis is disturbed leading to disruption of several metabolomic processes. In this way abnormal mineral homeostasis accelerates the development of cardiovascular disorders, which is supported by the fact that several studies have associated circulating calcium levels with vascular diseases and death. For example, changes in serum calcium levels can independently cause pathological disturbances in vascular tissue such as vascular calcification [37], coronary artery disease and myocardial infarction [38] and it can aggravate blood pressure and blood lipid levels, which are known risk factors of CVD. Studies have calculated an increase in the odds ratio of coronary heart disease and cardiovascular mortality when CaSR genetically varies due to single nucleotide polymorphisms and causes an increase in the circulating serum calcium [35]. Activation of CaSR by gadolinium in vitro in rat ventricular myocytes resulted in the phosphorylation of ERK1/2, JNK and p38 MAPK, the activation of the caspase 9 pathway and induction of apoptosis [39]. This observation was further supported by a study by Lu et al. showing that the disturbed extracellular calcium homeostasis also altered the calcium homeostasis in the Endoplasmic Reticulum (ER) and mitochondria eventually causing apoptosis and release of stress hormones which in turn can induce heart failure [40]. Several heterozygous inactivating mutations of CaSR have been shown to contribute to hypercalcemia, with high PTH values [5]. Nonetheless, familial hypocalciuric hypercalcemia (FHH) patients are found to have a unique variant of the CaSR gene even though they do not have hyperparathyroidism. This novel variant highlighted that serine 147 is an important site on the gene that is essential for sensing calcium as substitution of this site results in a disturbance in the calcium homeostasis [41]. On the other hand, gain-of-function mutations of CaSR have been shown to result in hypocalcaemia as well as hypomagnesemia [42]. Elevated levels of calcium in urine or hypercalciuria are also related to a variant of the CaSR gene, which decreases the expression of the receptor in renal tubules [43,44]. Everything considered, it has become clear that the sensing of calcium, being the main objective of the CaSR, is vital for the protection against cardiovascular disorders.

### 4.2. Heart Failure

Apoptosis is a type of cell death that occurs in cases of undesirable changes in the cell environment, such inflammation. Cardiomyocyte apoptosis, in response to a variety of stressors, contributes majorly to heart failure. This happens in a manner that enables cells to undergo apoptosis causing increased cardiac fibrosis and an increased number of hypertrophic cardiomyocytes [45]. Apoptosis and hypertrophy have been linked to the calcium homeostasis and CaSR in the vasculature. For example, CaSR activation leads to the activation of pathways that are calcium ion dependent, such as Ca^2+^/calmodulin-dependent protein kinase II (CaMKII) and calcineurin pathways [46], which are known to induce apoptosis and hypertrophy of cardiac myocytes. Furthermore, angiotensin II (AngII) has been observed to be involved in CaSR induced cardiac fibrosis in-vitro [47]. The presence of AngII increases the expression of CaSR in myocardial tissue which consecutively increases the proliferation of the fibroblasts, as well as their phenotypic transformation. This increased expression of CaSR additionally increases the intracellular calcium release, stimulates autophagy, and results in excess collagen formation and MEK pathway phosphorylation. When cardiomyocytes were injured with excess lipopolysaccharide (LPS) in-vitro, it was observed that CaSR is activated and that the activated CaSR acts via ER stress factors to induce autophagy and further alleviate apoptosis in these cells [48]. Furthermore, neutrophils that are recruited to sites of inflammation and in which CaSR is activated were observed to exacerbate MI by stimulating the apoptosis and myocardial fibrosis. The CaSR-activated neutrophils were seen to upregulate levels of interleukin-1 receptor (IL-1R), matrix metalloproteinase-2 (MMP2), alpha smooth muscle actin (α-SMA), collagen I and collagen III in cardiac fibroblasts via excessive release of IL-1β [49].

Cardiac fibrosis can result in excess matrix mineralization and collagen secretion which also stimulates proliferation and activation of these fibroblasts. There has been concrete proof that CaSR in expressed in cardiac fibroblasts [50]. Cardiac fibrosis is a very important aftermath in cardiomyopathy, along with cardiac remodeling and activation of inflammatory pathways [51]. One specific type of cardiomyopathy is diabetic cardiomyopathy. This is caused by diabetes mellitus, which is characterized by hyperglycemia due to a deficiency of insulin. Recently, it was shown that CaSR is one of the factors that modulates insulin secretion [52]. The constant hyperglycemic condition upregulates the expression of CaSR in cardiac fibroblasts which eventually leads to an increase in the intracellular calcium levels and activation of autophagy and the Smurf2 ubiquitin proteasome which is highly involved in protein degradation [52]. The increased levels of intracellular calcium in cardiac fibroblast also activate the TGF-β_1_/Smads pathway, which in turn stimulates the proliferation and myocardial fibrosis following activation of cardiac fibroblasts [27]. The above described studies demonstrate that CaSR can play important roles in the various pathological phases of heart failure.

### 4.3. Atherosclerosis and Vascular Calcification

Atherosclerosis is a chronic inflammatory disease, characterized by the progressive build-up of lipids and immune cells in the artery wall [53]. This pathology is initiated when ECs of the vessel wall are irritated by various stress factors and the subsequent accumulation of low-density lipoproteins (LDL) particles, leading to the chemotactic attraction of monocytes. These monocytes, after infiltrating the arterial wall, differentiate into macrophages and ingest cholesterol and store it in their cytoplasm to become ‘foam cells’ [54] triggering plaque development. During later stages of plaque development, SMCs from the medial layer undergo phenotypical changes and migrate into the intimal layer of the arteries to provide structural support to the growing lesion [55]. The formation and growth of atherosclerotic plaques can impede the blood flow either through vessel occlusion or plaque rupture, thereby causing other cardiovascular morbidities such as MI. Vascular calcification is a comorbidity in patients with hypertension, diabetes and kidney disease and a distinguishing characteristic of atherosclerosis. Vascular calcification associated with atherosclerosis is an active process that resembles bone formation regarding cellular processes [56]. Calcification is mainly caused by the accumulation of minerals in the medial layer of the vasculature wherein the SMCs ingest enormous amounts of minerals, mainly calcium, and are thereby reprogrammed to exhibit an osteo- and/or chondrogenic phenotype [56]. Calcification of aortic valves has also been recognized as an important cardiovascular risk factor in older adults. The valves of the aorta are calcified when valvular interstitial cells, which are the most abundant cells in valves, start expressing an osteoblast-like phenotype. Considering the notion that extracellular calcium is the main trigger for vascular calcification, it is not surprising that the expression of CaSR in interstitial cells is elevated in calcified aortic valves compared to healthy valves [57]. Downregulating CaSR in these cells has also been shown to reduce the calcium induced valvular calcification in-vitro. Valvular calcification is a predictor for valvular stenosis and eventually vascular calcification and atherosclerosis [58,59]. Even though the presence of CaSR in valvular interstitial cells has been discovered and the upregulated expression of the receptor in cases of calcification has been seen in-vitro and in-vivo, there is a lack of information on the exact mechanism by which CaSR influences stenosis in the valves. Atherosclerotic lesions are sources of calcium ions, activating CaSR in miscellaneous vascular cells that consequently stimulate the migration of immune cells to promote further plaque growth. Mineralization of the SMCs is demonstrated to be influenced by CaSR to a greater extent than by the movement of calcium ions via cell surface ion channels [20]. For example, a study demonstrated that overexpression of CaSR in vSMCs reduces calcium deposition by the cells and obstructs their change into a calcifying phenotype [60]. The same study also found that calcimimetics can be used to keep a constant expression of CaSR which inhibits calcification suggesting that this could be a good therapeutic strategy to combat vascular calcification. Further supporting this notion is the observation that increased concentrations of extracellular calcium in case of vascular calcification also activates the NLR family pyrin domain containing 3 (NLRP3) inflammasome and caspase-1 and increases levels of IL-1β in cardiac tissue in a CaSR dependent manner [61]. Moreover, vSMCs in atherosclerotic lesions also express MMP-2 which causes the breakdown of the extracellular matrix and thereby reduces plaque stability [62]. CaSR mediates the production of MMP-2 in the presence of cholesterol via the p13K/Akt pathway and can thereby also modulate plaque stability [63]. Furthermore, activation of CaSR in SMCs activates the MEK1/ERK1/2 pathways and the PLC-IP3 pathway, thereby increasing cell proliferation and survival respectively [64]. All of the in vitro and in vivo studies indicate that CaSR has a protective effect against atherosclerosis and vascular calcification by directly influencing and preventing mineral depositions in atherosclerotic plaques by SMCs.

### 4.4. Myocardial Infarction

The formation of atherosclerotic plaques can obstruct the blood flow in arteries, for example causing the blood flow to the cardiac muscles to be reduced, thereby reducing the oxygen levels. This diminishes the ability of the cardiac muscles to pump blood and causes abnormal cardiac rhythms and disequilibrium in myocardial metabolism causing a condition called myocardial ischemia [65]. Long-lasting ischemia results in the death of myocardial tissue and leads to the onset of the pathological condition termed MI [66]. The involvement of CaSR in myocardial ischemia is a debatable topic as it promotes ischemia and protects the myocardium against it. For example, the activation of CaSR exerts a protective function against endothelial injury in ischemia or reperfusion injury by inhibiting platelet activation [67]. Addititonaly, it also offers protection to cardiac tissues against apoptosis and oxidative stress caused by this pathological condition [68]. On the other hand, CaSR phosphorylates PKCδ, which is known to regulate intracellular calcium levels [69] by translocating it to the ER resulting in ER related stress hormones to further inducing apoptosis in the vascular cells in-vivo [27]. specifically ECs, the dysfunction of which is a critical factor in myocardial ischemia. ECs secrete inflammatory factors in excess to induce autophagy, apoptosis and ER stress on themselves and other vascular cells to further alleviate the injury [70]. An in vitro study on human umbilical vein endothelial cells (HUVECs) revealed that CaSR regulates the response of these cells to ischemia reperfusion injury. The expression of CaSR influences autophagy and angiogenesis to defer the cell migration and apoptosis, and can thereby play an important role in myocardial ischemia [71].

During MI, a severe inflammatory response is triggered and various immune cells are attracted to the site of inflammation and injury. It has been shown that CaSR not only responds to inflammation but can also further promote it [1]. For example, activation of CaSR in T lymphocytes was shown to result in an increased release of IL-6 and TNFα as cellular responses to stress signals into circulation via MAPK-ERK, MAPK-JNK, and NF-κB [27,29]. This activation causes further deterioration and thereby aggravates the pathological consequences of MI. This is further supported by the observation that silencing of CaSR and thereby blocking of the NF-κB pathway in T lymphocytes reduces the apoptosis rate and the secretion of pro-inflammatory cytokines (Th1 cytokines) and anti-inflammatory cytokines (Th2 cytokines) [29]. Further linking CaSR with MI, is the notion that both CaSR and NLRP3 inflammasome are activated in the peripheral neutrophils upon MI occurrence [49]. Neutrophils play an important role in the onset of MI by releasing various pro-inflammatory cytokines and chemokines. Furthermore, it could be shown that the CaSR-activated neutrophils instigate cardiomyocyte apoptosis and thereby aggravate ischemia [49]. Besides neutrophils, macrophages also play an important role in MI. It has been observed that the activation of CaSR promotes the secretion of IL-1β and TNF-α in monocyte-derived macrophages [27,72]. Furthermore, blocking of CaSR in macrophages attenuates micropinocytosis, which interferes with the ability of the cells to sense nucleotide-binding oligomerization domain-containing protein 2 (NOD2) ligands and delivery of ligands to Toll-like receptors (TLRs) [32]. In summary, CaSR seems to play an important role in ischemia induced MI by influencing various cells and processes.

### 4.5. Hypertension

Persistent elevation of blood pressure in arteries is a medical condition called hypertension, an extremely important cause of heart failure. Hypertension causes organ failure leading to vascular remodeling, which is distinguished by apoptosis, inflammation and fibrosis [73]. The pathogenesis of hypertension is also promoted by vascular inflammation. Intake of appropriate amounts of dietary calcium is necessary to provide a protective effect on cardiovascular tissues against hypertension. Various studies have already demonstrated that the presence of CaSR in vSMCs is necessary to keep the blood pressure level within the physiological range and to reduce hypertension [22,74]. This influence on blood pressure by CaSR can be explained by its inhibitory effects on the renin-angiotensin-aldosterone system (RAAS) [75]. For example, systemic inhibition of CaSR using a known allosteric inhibitor NPS-2143 results in an elevated blood pressure in normotensive rats [76]. RAAS works through PLC-IP3 pathways, and the unwarranted activation of RAAS can pathologically cause excessive proliferation of vSMCs, vasoconstriction and myocardial hypertrophy [27]. Inhibition of CaSR therefore increases the proliferation of vSMCs and reduces apoptosis. A study conducted on rats with spontaneous hypertension indicated that CaSR is selectively activated in the myocardium triggering reverse cardiac remodeling [77]. Along with hypertension and increasing age, the rats in this study also showed a progressive increase of myocardial apoptosis, hypertrophy and a deteriorating cardiac function. However, the activation of CaSR in these rats led to a reduction in blood pressure, suggesting that CaSR plays a role in inhibiting the progression of myocardial injuries through the inhibition of RAAS in cardiac tissues [77]. In line with this, another study demonstrated that in the mesenteric arteries of hypertensive rats, the expression of CaSR is decreased [27]. This reduced expression of CaSR enhanced the vasodilation of the arteries and regulated the vascular tension and the blood pressure in the rats through the PLC-IP3/cAMP/RAAS pathways [27]. In rats with spontaneous hypertension, AngII also activates CaSR, as well as NLRP3 inflammasome in vSMCs [78].

Vasorelaxations and constrictions are the main regulators of blood pressure in the vascular system. The vascular tension is controlled by intracellular calcium in vascular tissue [79]. Increased levels of calcium stimulate CaSR in the endothelium which promotes endothelium dependent vasorelaxation through an increase in the release of ^●^NO via the eNOS pathway [26]. CaSR influences the ^●^NO system via heterodimeric TRPV4-TRPC1-mediated calcium influx [80]. In the vascular microenvironment, different pumps act together to influx calcium ions in excess to form what are called calcium clouds. These calcium clouds trigger the activation of CaSR in ECs. Vasorelaxations are aided by hyperpolarization of vSMCs that are caused by changes in ion channels. However, the activation of these channels is also supported by the stimulation of CaSR, suggesting the existence of a prominent feedback loop [81].

In conclusion, it is believed that CaSR responds to systemic and local calcium ion concentrations to modulate the vascular tone of the vessels [14]. The normal expression of CaSR thereby functions as a protective shield against hypertension and its associated cardiac pathologies.

### 4.6. Obesity’s Influence

In recent years, obesity has progressed into a disease that torments a large portion of the population worldwide. Obesity is a condition in which there is an excess accumulation of lipids in the body resulting in increased and enlarged adipose tissue. CVDs linked to obesity are mostly due to adipose tissue dysfunction, its enlargement, and the release of inflammatory factors from the adipose tissue [82]. Enlarged adipose tissues also show higher expression levels of CaSR [83]. It has been observed that the activation of CaSR in human preadipocytes stimulates autophagy, which is an essential mechanism for cellular homeostasis, because when damaged organelles are not degraded by lysosomes this leads to cardiometabolic comorbidities especially in conditions of obesity [84]. All of the studies on adipose tissue autophagy unfortunately do not stipulate the exact cell type is responsible for this process and therefore more research is needed to elucidate this further [85]. CaSR is believed to activate the NLRP3 inflammasome via the ERK pathway in preadipocyte, as in many other cell types [86]. In cases of obesity, the adipose tissue expands and starts accumulating immune cells to facilitate the already disturbed metabolism [87]. Activating CaSR in macrophages, which are prominent in adipose tissues, induces inflammation in preadipocytes via the NLRP3 inflammasome pathway suggesting that interfering with the expression of CaSR in obesity could work as a therapeutic mechanism. Looking at all the studies focusing on the roles of CaSR in adipose tissue and obesity, it could be inferred that CaSR does not contribute towards protection of the adipose tissue, but it rather promotes a dysfunctional state and could thereby contribute to CVDs [88].

## 5. Presence and Function of CaSR in Other Organs and Organ Systems

Besides its important direct role in cardiovascular related cells, CaSR is also known to be present on the surface of different cells and to influence plasma mineral ion concentrations, especially calcium and phosphate [89] (Figure 3). The predominant expression of CaSR is in the parathyroid glands. Even though the well-known role of the receptor is the negative control of parathyroid function to influence the calcium homeostasis via the suppression of parathyroid hormone (PTH) secretion [4,90], the receptor has now been studied in various other contexts as well leading to an increased understanding of the role this receptor can play in various (patho)physiological conditions. To start with, the parathyroid CaSR is also involved in monitoring the plasma levels of L-amino acids [91]. Furthermore, CaSR is also expressed in thyroidal C-cells, where it regulates calcitonin secretion to again influence mineral homeostasis in the blood [92]. Another tissue in which a rather high expression of CaSR is seen is the kidney [93], wherein the receptor influences the mineral homeostasis by influencing the resorption and excretion of ions in urine. The expression of CaSR along the nephron influences the final compositions of urine and plasma that leaves the renal area [94]. The activated renal CaSR is also involved in inhibiting the tubular response to vasopressin thereby limiting water resorption by the kidneys [95]. CaSR is also expressed in osteoblasts, osteocytes, osteoclasts, and chondrocytes [96] and along with calcium ions the receptor is involved in bone development and maintenance [27,97]. In addition, CaSR is believed to be widely expressed in the nervous system [98]. Here, the receptor is part of processes including the ion channel metabolism, release of neurotransmitters, action potential initiation, and polarization and depolarization of neurons [99]. CaSR is similarly found to be expressed in the gut epithelium, aiding with the barrier function [100] and related cellular functions there, which include the maintenance of an equilibrium of the gut microbiota and immune cell chemotaxis. The importance of CaSR in this organ is further supported by the observation that the absence of CaSR in the gut resulted in increased local and systemic inflammation in mouse models [101]. The airways contain endothelial and smooth muscle cells that also express CaSR. In cases of severe asthma, the levels of cationic proteins and polyamines in the airway increases exponentially and it is believed that CaSR is implicated in this [102]. The receptor promotes disease development by aggravating the hyperresponsiveness, constriction and inflammation of the airway. Finally, around a decade ago CaSR expression was also demonstrated in adipocytes and pre-adipocytes [103]. The expression of the receptor in these cells is involved in inflammatory responses of the adipose tissue and has an anti-lipolysis function [104,105].

## 6. Concluding Remarks

CaSR is expressed on the surfaces of cells of a wide variety of tissues, where it performs numerous functions by means of activating diverse signaling pathways depending on the bound ligand. CaSR in general plays an important role in various pathological processes, such as cardiovascular pathologies. The receptor influences and promotes acute and chronic inflammation and triggers various intracellular signaling pathways, thereby participating in atherosclerosis, vascular calcification, cardiomyopathy, cardiac fibrosis, and myocardial infarction. CaSR is implicated in various diseases, making it a very interesting pharmacological target to study. Although studies on targeting of this receptor to inhibit the progression of CVDs are scarce, targeting of this receptor in vascular tissues could be an interesting novel therapeutic mechanism to control and manipulate the cardiovascular biology. Therefore, future studies should focus on the therapeutic potential of the CaSR in inflammatory processes and cardiovascular pathologies to elucidate the exact potential.

## Figures and Tables

**Figure 1 ijms-22-02478-f001:**
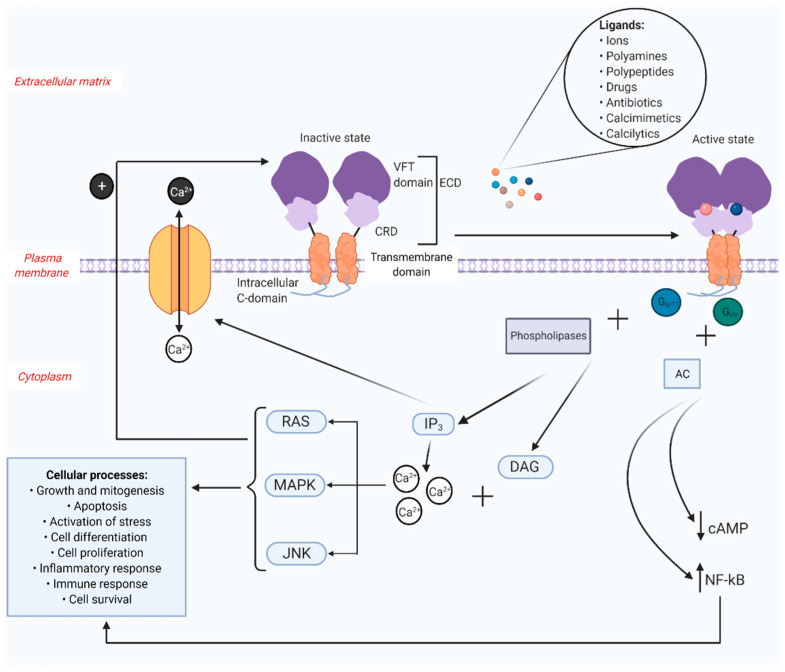
Structure and function of CaSR. The CaSR is composed of three structural domains: the extracellular domain (ECD) consisting of a Venus flytrap (VFT) domain and a cysteine rich domain (CRD); the transmembrane domain connected by the CRD to the VFT; and an intracellular C-domain. The extracellular ligands of CaSR include various ions, polyamines, polypeptides, drugs, antibiotics, calcimimetics and calcilytics. These ligands bind to the inactive receptor to initiate conformational changes to the receptor resulting in receptor activation. The active receptor differs from the inactive receptor in such a way that the protomers in the VFT are in close proximity to each other. The surface receptor, when activated by its ligands, can trigger numerous intracellular signaling pathways, which in turn activate different cellular processes. The activation of CaSR couples G_i/o_ and G_q/11_ families of heterotrimeric G proteins. Along with phospholipases, G_q/11_ prompts inositol triphosphate (IP_3_) and diacylglycerol (DAG) pathways. IP_3_ controls calcium channels on the surface of the cell, thereby regulating the efflux and influx of calcium ions. DAG, along with the cytosolic calcium will activate rat sarcoma (RAS)/mitogen-activated protein kinase (MAPK)/c-Jun n-terminal kinases (JNK) pathways which are involved in the initiation of several cellular processes. These pathways also engage in a feedback mechanism required for the proper functioning of the receptor. G_i/o_ interacts with adenylate cyclase (AC) to curb the production of cyclic adenosine monophosphate (cAMP). The receptor also upregulates the transcription factor nuclear factor kappa-light-chain-enhancer of activated B cells (NF-kB), which is again associated with various cellular processes (Created with BioRender.com).

**Figure 2 ijms-22-02478-f002:**
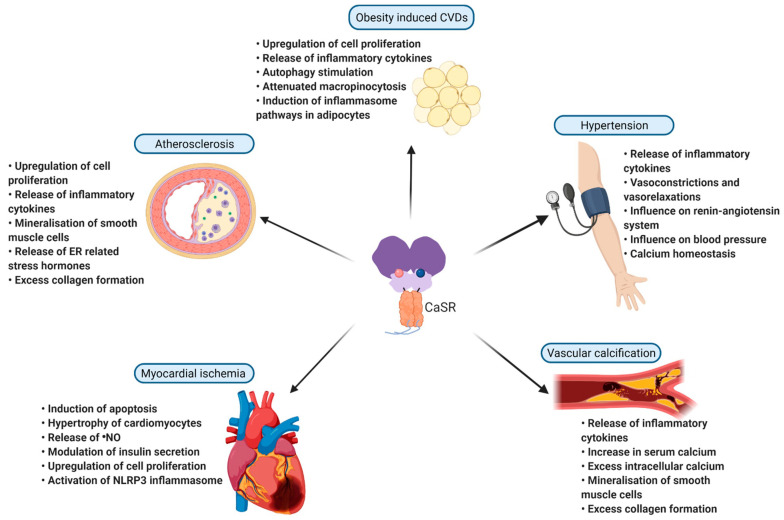
CaSR in CVDs. CaSR is implicated in various cardiovascular disorders, where it exerts its role by influencing various factors involved in the development and progression of the disease. CaSR is involved in myocardial ischemia, vascular calcification, hypertension, atherosclerosis, and obesity induced CVDs. This figure gives a brief summary of the involvement of CaSR in the above-mentioned CVDs and the ways in which CaSR influences each one of them (created with BioRender.com). ER: endoplasmic reticulum; NLRP3: NLR family pyrin domain containing 3; ^●^NO: nitric oxide.

**Figure 3 ijms-22-02478-f003:**
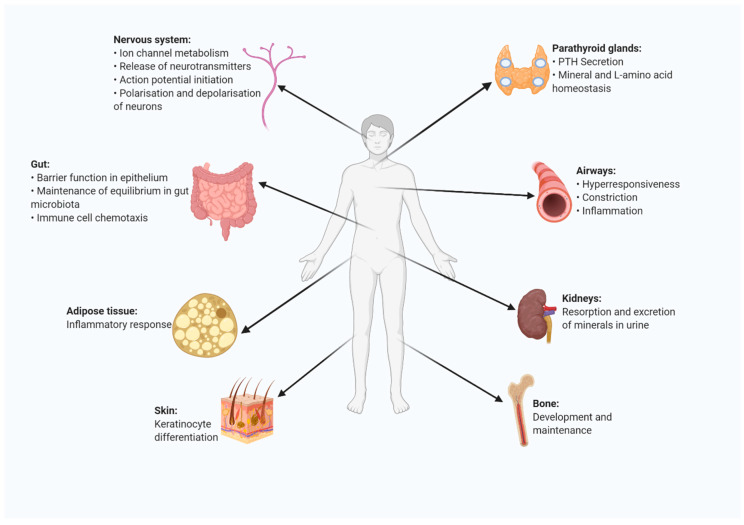
CaSR expression and function in a range of organs and tissues in the human body. CaSR plays a (patho)physiological role in multiple organs and tissues in humans. CaSR is present in parathyroid glands, kidneys, bone, skin, the gut, airways, adipose tissues and in the nervous system, where it exerts various functions (created with BioRender.com).

## Data Availability

Not applicable.

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
