# Peer review of "Calcium-Sensing Receptor (CaSR), Its Impact on Inflammation and the Consequences on Cardiovascular Health"

_ijms, 2021, doi:10.3390/ijms22052478_

Round 1
Reviewer 1 Report
The work set up here requires a capable mind to fully integrate the data obtained on this subject. On the other hand, the work does not give that much focus on cardiovascular, but more on inflammatory states addressed by these receptors, which on the other hand have CVD effect.
For example, in myocardial infarction the effect of CaR was more discussed regarding inflammation than the natural impact of calcium levels and MI.
Association of Genetic Variants Related to Serum Calcium Levels With Coronary Artery Disease and Myocardial Infarction.
Larsson SC, Burgess S, Michaëlsson K
JAMA. 2017 Jul 25; 318(4):371-380
Alanine to serine polymorphism at position 986 of the calcium-sensing receptor associated with coronary heart disease, myocardial infarction, all-cause, and cardiovascular mortality.
März W, Seelhorst U, Wellnitz B, Tiran B, Obermayer-Pietsch B, Renner W, Boehm BO, Ritz E, Hoffmann MM
J Clin Endocrinol Metab. 2007 Jun; 92(6):2363-9.
Also the role of these receptors on vasopressin if far neglected as is the role of vasopressin for example on heart failure.
Endocr Metab Immune Disord Drug Targets
. 2018;18(5):458-465. doi: 10.2174/1871530318666180212095235.
Vasopressin in Heart Failure
Michele Iovino, Massimo Iacoviello, Giovanni De Pergola, Brunella Licchelli, Emanuela Iovino 1, Edoardo Guastamacchia, Vito A Giagulli, Vincenzo Triggiani
I do not disagree with the approach on inflammation and it is a sound approach, but it is not what is addressed in the abstract or what we are looking when starting the work. It seems more of the role of CaR on inflammation and its consequences on CVD level. Even obesity is an pro-inflammatory and that is the organization of the work. The authors need to clarify what they want up front. Also the introductory section is too broad.
Minor issues:
Figure 1 and 2 (and 3) do not have much quality. Need to be improved. (quality of the image but also NF-ĸB)
I am a huge fan of large introductions, where I learn a great deal, but it is quite too much for me that only at page 6 and an half do the issues of this review (cardiovascular) are beginning to be address. The authors should be more focused.
Correct typo and some inconsistences:
‘The structure of the inactive as well as active receptor have’
‘an L-amino acid binds to receptor’ an L-amino acid binds to the receptor
‘various cellular processes mentioned earlier’; where? In the figure? In the text at that point, it was never mentioned
‘cell type in cardiovascular tissues’ to ‘cell type in vascular tissues’
‘nitric oxide (NO)’ to nitric oxide (●NO)
Author Response
Response to reviewer 1
The work set up here requires a capable mind to fully integrate the data obtained on this subject. On the other hand, the work does not give that much focus on cardiovascular, but more on inflammatory states addressed by these receptors, which on the other hand have CVD effect. For example, in myocardial infarction the effect of CaR was more discussed regarding inflammation than the natural impact of calcium levels and MI.
Association of Genetic Variants Related to Serum Calcium Levels With Coronary Artery Disease and Myocardial Infarction.
Larsson SC, Burgess S, Michaëlsson K
JAMA. 2017 Jul 25; 318(4):371-380
Alanine to serine polymorphism at position 986 of the calcium-sensing receptor associated with coronary heart disease, myocardial infarction, all-cause, and cardiovascular mortality.
März W, Seelhorst U, Wellnitz B, Tiran B, Obermayer-Pietsch B, Renner W, Boehm BO, Ritz E, Hoffmann MM
J Clin Endocrinol Metab. 2007 Jun; 92(6):2363-9.
We would like to thank the reviewer for his/hers critical evaluation of our manuscript. We agree that a large focus is on inflammatory states/processes, which obviously have large effects on the cardiovascular system. We have therefore changed the title of the our review and specified this in the abstract, introduction and conclusion. We have also added the literature specified by the reviewer to the manuscript.
Also the role of these receptors on vasopressin if far neglected as is the role of vasopressin for example on heart failure.
Endocr Metab Immune Disord Drug Targets.
2018;18(5):458-465. doi: 10.2174/1871530318666180212095235.
Vasopressin in Heart Failure
Michele Iovino, Massimo Iacoviello, Giovanni De Pergola, Brunella Licchelli, Emanuela Iovino 1, Edoardo Guastamacchia, Vito A Giagulli, Vincenzo Triggiani
We understand that our review does not cover the role of vasopressin in heart failure. Unfortunately, there is no literature yet available to describe the precise role of CaSR with heart failure mediated by vasopressin and therefore we have omitted these publications from our review.
I do not disagree with the approach on inflammation and it is a sound approach, but it is not what is addressed in the abstract or what we are looking when starting the work. It seems more of the role of CaR on inflammation and its consequences on CVD level. Even obesity is an pro-inflammatory and that is the organization of the work. The authors need to clarify what they want up front. Also the introductory section is too broad.
We agree with the reviewer and have now revised our manuscript, especially title, abstract and introduction to stress that the main focus is on inflammation and its consequence on CVD (rather than primarily CVD). We have also restructured the introduction section to increase the readability.
Minor issues:
Figure 1 and 2 (and 3) do not have much quality. Need to be improved. (quality of the image but also NF-ĸB)
Thank you for noticing this, we have adjusted everything accordingly. In order to improve the quality and shorten the introduction section we have now combined original figure 1 and 2.
I am a huge fan of large introductions, where I learn a great deal, but it is quite too much for me that only at page 6 and an half do the issues of this review (cardiovascular) are beginning to be address. The authors should be more focused.
We appreciate this suggestion, and we have significantly reconstructed the review according to the suggestion.
Correct typo and some inconsistences:
‘The structure of the inactive as well as active receptor have’
‘an L-amino acid binds to receptor’ an L-amino acid binds to the receptor
‘various cellular processes mentioned earlier’; where? In the figure? In the text at that point, it was never mentioned
‘cell type in cardiovascular tissues’ to ‘cell type in vascular tissues’
‘nitric oxide (NO)’ to nitric oxide (●NO)
Thank you for noticing these minor points, we have adjusted everything accordingly
Reviewer 2 Report
In a manuscirpt by Sundararaman et al. reviewed the role of calcium-sensing receptor in cardiovascular system and disorders. I have read the manuscript with much interest and have an impression that the review provided by the authors is very detailed and covers all aspects of pathophysiology related to CVS. How about association of CSR with calcific valve disorders esp aortic stenosis? Could the authors refer to some data? Gap in the evidence paragraph would be very useful to broadcast the tone for future research perspective and possibilities.
Author Response
Response to reviewer 2
In a manuscript by Sundararaman et al. reviewed the role of calcium-sensing receptor in cardiovascular system and disorders. I have read the manuscript with much interest and have an impression that the review provided by the authors is very detailed and covers all aspects of pathophysiology related to CVS. How about association of CSR with calcific valve disorders esp aortic stenosis? Could the authors refer to some data? Gap in the evidence paragraph would be very useful to broadcast the tone for future research perspective and possibilities.
We would like to thank the reviewer for his/hers positive evaluation of our manuscript. We have now incorporated the suggestions provided by the reviewer (lines 295-299).
Round 2
Reviewer 1 Report
The authors responded to my questions. I recommend acceptance.